# `C2-DPO`: Constrained Controlled Direct Preference Optimization

**Kavosh Asadi** *kavasadi@amazon.com*
*Amazon*
**Xingzi Xu** * *xingzixu@amazon.com*
*Amazon*
**Julien Han** *hameng@amazon.com*
*Amazon*
**Ege Beyazit** *beyazit@amazon.com*
*Amazon*
**Idan Pipano** *idan.pipano@campus.technion.ac.il*
*Technion*
**Dominique Perrault-Joncas** *joncas@amazon.com*
*Amazon*
**Shoham Sabach** *ssabach@amazon.com*
*Amazon*
**Mohammad Ghavamzadeh** *ghavamza@amazon.com*
*Amazon*
**Karim Bouyarmane** *bouykari@amazon.com*
*Amazon*

**Reviewed on OpenReview:** *https://openreview.net/forum?id=7h5Ho9t5NL*

## Abstract

Direct preference optimization (`DPO`) has emerged as a promising approach for solving the alignment problem in AI. In this paper, we make two counter-intuitive observations about `DPO`. First, we show that the `DPO` loss could be derived by starting from an alternative optimization problem that only defines the KL guardrail on in-sample responses, unlike the original RLHF problem where guardrails are defined on the entire distribution. Second, we prove a surprising property of this alternative optimization problem, where both the preferred and rejected responses tend to decrease in probability under its optimal policy, a phenomenon typically displayed by DPO in practice. To control this behavior, we propose a set of constraints designed to limit the displacement of probability mass between the preferred and rejected responses in the reference and target policies. The resulting algorithm, which we call Constrained Controlled DPO (`C2-DPO`), has a meaningful RLHF interpretation. By hedging against the displacement, `C2-DPO` provides practical improvements over vanilla `DPO` when aligning several language models using standard preference datasets.

## 1 Introduction

Ensuring AI systems act in accordance with human preferences, also known as the *alignment* problem, has become a critical focus in machine learning. Reinforcement Learning from Human Feedback (RLHF) has emerged as one promising approach (Christiano et al., 2017). RLHF proceeds by first learning a reward model (RM), and then employing standard RL algorithms to maximize the RM while keeping the model close to a reference model. Recent years have witnessed the emergence of algorithms that solve the two RLHF sub-problems in a single step, chief among them being the Direct Preference Optimization (`DPO`) algorithm (Rafailov et al., 2023). `DPO` proceeds by leveraging the closed-form solution of the RLHF objective

---

*Corresponding author.

and using the preference dataset to align the model, thus bypassing the explicit reward-learning and the need to sample new responses during training. Since then, numerous extensions and successors have been proposed, e.g., `IPO` (Azar et al., 2024) and `CDPO` (Mitchell, 2024), underscoring the need for a deeper investigation into this emerging class of algorithms to connect the underlying principles.

In this paper, we start with the counter-intuitive observation that the `DPO` loss can be derived from an alternative optimization problem that imposes the KL penalty only on in-sample responses - those present in the preference dataset - rather than on the full output distribution, as done in traditional RLHF. We show that this subtle shift has a significant implication: the alternative optimization problem incentivizes in-sample probability reduction in DPO. We formally prove that under the optimal solution to this new problem, both the preferred and rejected responses tend to decrease in probability. This phenomenon, while counter-intuitive, mirrors recent findings about DPO behavior (e.g., Adler et al. 2024; Xu et al. 2024; Pal et al. 2024; Fisch et al. 2024; Xiao et al. 2024; Shen et al. 2024; Wu et al. 2024; D'Oosterlinck et al. 2024; Pang et al. 2024), and is referred to as *likelihood displacement* by Razin et al. (2025). We then show that the above in-sample probability reduction phenomenon is *shared among DPO extensions/successors* by developing a simple classification framework that unifies the family of DPO-style algorithms.

Leaning on these insights, we propose a family of constraints that provably control likelihood displacement in DPO-style algorithms. The constraints are designed to limit the movement of winner-loser probability mass between the reference and target policies. Our proposed algorithm, Constrained Controlled DPO (`C2-DPO`), optimizes the `DPO` objective under these constraints, has a meaningful RLHF interpretation, and requires no extra computation. We evaluate the effectiveness of our constraints in enhancing preference alignment across two datasets and three models with up to 13B parameters, and show that `C2-DPO` outperforms vanilla `DPO` and several other baselines, delivering higher-quality final models when assessed holistically on the standard MT-Bench dataset (Zheng et al., 2023).

## 2 Preliminaries

We present the key ingredients of preference optimization on which we will build in the subsequent sections. In this setting, we are given a dataset $\mathcal{D}$ of triplets $(x, y_w, y_l)$, where $x$ is a prompt, while $y_w$ and $y_l$ reflect our preference in choosing response $y_w$ over $y_l$ conditioned on $x$. We are also given a reference policy $\pi_{\mathrm{ref}}$ (often the SFT checkpoint $\pi_{\mathrm{SFT}}$) which serves as a guardrail.

In RLHF, we first employ $\mathcal{D}$ to train a parameterized RM, $r_\phi$, and then use it to solve the following:

$$\max_\theta \ \mathbb{E}_x\Big[\mathbb{E}_{y\sim\pi_\theta}\big[r_\phi(x,y)\big] - \beta\mathbb{KL}\big(\pi_\theta(\cdot|x)||\pi_{\mathrm{ref}}(\cdot|x)\big)\Big], \tag{1}$$

where $\beta > 0$ is a hyper-parameter denoting the relative importance of reward maximization against ensuring a low deviation from $\pi_{\mathrm{ref}}$. The RM is learned by minimizing the cross-entropy (CE) loss:

$$\min_\phi \sum_{(x,y_w,y_l)\in\mathcal{D}} -\log\sigma\big(r_\phi(x,y_w) - r_\phi(x,y_l)\big), \tag{2}$$

assuming that preferences follow the Bradley-Terry (BT) model: $p(y_w \succ y_l \mid x) = \sigma\big(r(x,y_w) - r(x,y_l)\big)$ where $\sigma(x) = 1/(1+\exp(-x))$ is the sigmoid function and $r$ is the latent reward of the annotator. Fine-tuning $\pi_\theta$ in the RLHF approach splits into two stages: reward-learning using the BT model, followed by a policy optimization using equation 1. More recently, an emerging family of algorithms solve the above two problems in a single stage: Direct Preference Optimization (DPO)-style algorithms. The loss function of `DPO` is derived from the RLHF problem equation 1 using the recipe from (Rafailov et al., 2023). The key insight here is that problem equation 1 admits the following closed-form solution: $\pi^*(y|x) = \pi_{\mathrm{ref}}(y|x)\exp\big(r(x,y)/\beta\big)/Z(x)$ with $Z(x)$ as the partition function. We can rewrite this as

$$r(x,y) = \beta\log\frac{Z(x)\pi^*(y|x)}{\pi_{\mathrm{ref}}(y|x)} \ . \tag{3}$$

Substituting $r(x,y)$ from equation 3 into equation 2, the partition function $Z(x)$ cancels out, leading to the optimization problem solved by `DPO`:

$$\min_{\theta} \sum_{(x,y_w,y_l)\in\mathcal{D}} -\log\sigma\left(\beta\log\frac{\pi_\theta(y_w|x)}{\pi_{\text{ref}}(y_w|x)} - \beta\log\frac{\pi_\theta(y_l|x)}{\pi_{\text{ref}}(y_l|x)}\right) . \tag{4}$$

## 3 KL in DPO: Implicit Guardrailing with a Counter-intuitive Side Effect

Consider the optimization problem formed by replacing the KL-penalty in equation 1 with one that only operates on in-sample responses $S_x = y_w, y_l \mid (x, y_w, y_l) \in \mathcal{D}$:

$$\max_{\theta} \ \mathbb{E}_x\left[\mathbb{E}_{y\sim\pi_\theta}\left[r_\phi(x,y)\right] - \beta\sum_{y\in S_x}\pi_\theta(y|x)\log\frac{\pi_\theta(y|x)}{\pi_{\text{ref}}(y|x)}\right] . \tag{5}$$

Comparing to equation 1, recall that $\mathbb{KL}\big(\pi_\theta(\cdot|x)||\pi_{\text{ref}}(\cdot|x)\big) := \sum_y \pi_\theta(y|x)\log\frac{\pi_\theta(y|x)}{\pi_{\text{ref}}(y|x)}$ sums over all $y$, whereas equation 5 sums only over $S_x$. At first glance, the two optimization problems look quite similar, but finding the closed-form solution of equation 5 requires delicate analysis since the penalty only sums over $S_x$, leading to more involved KKT optimality conditions. Via this KKT analysis (details in Appendix A), problem equation 5 admits a closed-form solution:

**Remark 3.1** (Explicit form of the solution). *The optimal policy for equation 5 takes one of two forms depending on whether out-of-sample responses have higher rewards than in-sample ones. Defining $\hat{r}_x :=$ $\max_{y'\notin S_x} r_\phi(x,y')$:*

*Case I: If $\hat{r}_x > \beta\log\sum_{y\in S_x}\pi_{ref}(y|x)e^{\frac{r_\phi(x,y)}{\beta-1}}$, then*

$$\begin{cases} \pi_\theta^*(y|x) = \pi_{ref}(y|x)e^{(r_\phi(x,y)-\hat{r}_x)/\beta-1} & y\in S_x; \\ \pi_\theta^*(y|x) = 0 & y\notin S_x, r_\phi(x,y) < \hat{r}_x; \\ \sum_{y\notin S_x, r_\phi(x,y)=\hat{r}}\pi_\theta^*(y|x) = 1 - \sum_{y\in S_x}\pi_{ref}(y|x)e^{(r_\phi(x,y)-\hat{r}_x)/\beta-1} & y\notin S_x, r_\phi(x,y)=\hat{r}_x. \end{cases} \tag{6}$$

*Case II: Otherwise,*

$$\begin{cases} \pi_\theta^*(y|x) = \frac{\pi_{ref}(y|x)e^{r_\phi(x,y)/\beta}}{\sum_{y'\in S_x}\pi_{ref}(y'|x)e^{r_\phi(x,y')/\beta}} & y\in S_x; \\ \pi_\theta^*(y|x) = 0 & y\notin S_x. \end{cases} \tag{7}$$

$\square$

Despite this complex form, both cases satisfy the key property that for any $y, y' \in S_x$,

$$r_\phi(x,y) - r_\phi(x,y') = \beta\log\frac{\pi_\theta(y|x)}{\pi_{\text{ref}}(y|x)} - \beta\log\frac{\pi_\theta(y'|x)}{\pi_{\text{ref}}(y'|x)}. \tag{8}$$

We further prove that substituting the explicit solutions into the Bradley-Terry model recovers the standard DPO loss equation 4:

**Lemma 3.1.** *Problem equation 5 yields the same `DPO` loss as problem equation 1.*

At first glance, problems equation 1 and equation 5 appear similar, as both maximize expected reward with a KL-based penalty. However, Lemma 3.1 reveals that DPO can equivalently be derived from the latter, where regularization applies only to the small subset $S_x$ rather than the full response space. This has important implications for understanding DPO's behavior. We note that the introduced penalty in equation 5 is not a proper divergence, in the sense that it could be negative. Thus, the Lemma 3.1 could be viewed as evidence that when we move from RLHF to `DPO`, we no longer explicitly enforce the $\mathbb{KL}$ penalty. However, as using larger values of $\beta$ results in smaller KL deviations during `DPO` training (see Figure 1-Left), we can still empirically use KL regularization as an effective guardrail. This arguably makes sense, because in DPO there is little incentive for the model to shift probability mass for responses that are quite different from in-sample responses, and explicit out-of-sample guardrailing may not be necessary.

While this minimal guardrailing is effective, it leads to counter-intuitive behavior. The penalty term in equation 5 can be negative, unlike the standard KL divergence, which is always non-negative. More importantly,

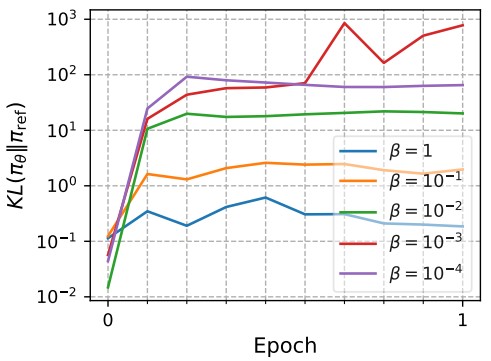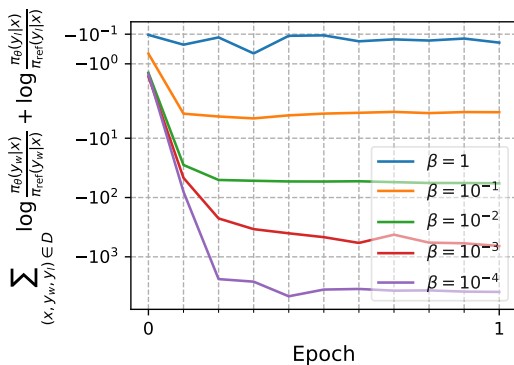

Figure 1: **Left:** Increasing $\beta$ in DPO leads to effective out-of-sample guardrailing. We ran DPO with different values of $\beta$ starting from the Zephyr-7B initial checkpoint on the UltraFeedback dataset. We then estimated the KL divergence between $\pi_\theta$ and $\pi_{\text{ref}}$ by autoregressively sampling $N = 32$ responses from $\pi_\theta$ for each prompt in the test set, followed by computing $\frac{1}{N} \sum_{i=1}^{N} \log \frac{\pi_\theta(y_i|x)}{\pi_{\text{ref}}(y_i|x)}$ and averaging over prompts. **Right:** Reduction of in-sample probabilities in DPO training.

the overall objective in equation 5 can increase by simultaneously decreasing the probabilities of both winner and loser responses (making $\log \pi_\theta(y \mid x)/\pi_{\text{ref}}(y \mid x) < 0$ for all $y \in S_x$). In contrast, this behavior is not incentivized in the standard RLHF problem equation 1. There, the KL penalty is non-negative by defini-tion, so reducing in-sample probabilities would require increasing out-of-sample probabilities correspond-ingly, providing no net benefit. We now formalize this observation:

**Lemma 3.2.** *Consider pairs $(x, y)$ with $y \in S_x$. If $r_\phi(x, y) \leq \max_{y' \notin S_x} r_\phi(x, y')$, then any optimal solution to equation 5 satisfies $\pi_\theta(y|x) \leq e^{-1} \pi_{ref}(y|x)$.*

The proof (Appendix A) follows from the subset-constrained penalty structure in equation 5. This result does not hold for standard RLHF equation 1, where the full KL penalty prevents this counter-intuitive sys-tematic probability reduction behavior. The lemma shows that in-sample responses—whether preferred or rejected—experience probability reduction whenever their rewards are dominated by out-of-sample rewards. Since $|S_x|$ is typically very small relative to the full response space (in the extreme case, a single preferred and rejected response), this condition is likely to hold frequently, explaining the widespread likelihood dis-placement observed in practice.

Recent work has observed that during DPO training, all in-sample probabilities—including those of preferred responses—tend to decrease (e.g., Adler et al. 2024; Xu et al. 2024; Pal et al. 2024; Fisch et al. 2024; Xiao et al. 2024; Liu et al. 2024; Yin et al. 2024; Guo et al. 2024; Yuzi et al. 2025; Razin et al. 2025; Deng et al. 2025; Xiliang et al. 2025; Huang et al. 2025; Yunan et al. 2025; Pang et al. 2024). Figure1 (Right) confirms this trend: the sum of log-ratios $\log \frac{\pi_\theta(y_w|x)}{\pi_{\text{ref}}(y_w|x)} + \log \frac{\pi_\theta(y_l|x)}{\pi_{\text{ref}}(y_l|x)}$ decreases substantially during training. Lemma 3.2 hints at the underlying reason for this counter-intuitive behavior, which has been referred to as *likelihood displacement* of in-sample probabilities (Razin et al., 2025). To the best of our knowledge, while this has been reported in previous empirical studies, Lemma 3.2 is among very few theoretical results explaining this counter-intuitive phenomenon.

We conclude this section by noting that we proved in-sample probability reduction for DPO alone, and so it would be natural to ask if some of its main successors, e.g., `IPO` (Azar et al., 2024) and `CDPO` (Mitchell, 2024), share the same property. In the next section, we answer this question affirmatively by developing a classification framework that unifies the family of DPO-style algorithms.

## 4 A Classification View of DPO-style Algorithms

We now show that DPO-style algorithms can be interpreted as classification methods, where the objective is defined solely over in-sample responses. As a result, similar to DPO, none of these algorithms applies any

direct guardrailing on out-of-sample responses. Recall that the standard classification setting has three main ingredients.

**First**, we construct a hypothesis space by defining probabilities assigned to each class. In DPO-style algorithms, these probabilities are implicitly defined as:

$$p_\theta(x, y_w, y_l) := \text{softmax}\big(r_\theta(x, y_w), r_\theta(x, y_l)\big) \ , \tag{9}$$

where $r_\theta$ is the reward defined in equation 3 having substituted $\pi^*$ with $\pi_\theta$. Under $p_\theta$, the probability assigned to the *winner (preferred)* response $y_w$, denoted by $p_\theta^w$, does not depend on the partition function $Z(x)$ and can be written as:

$$p_\theta^w(x, y_w, y_l) := \frac{\big(\frac{\pi_\theta(y_w|x)}{\pi_{\text{ref}}(y_w|x)}\big)^\beta}{\big(\frac{\pi_\theta(y_w|x)}{\pi_{\text{ref}}(y_w|x)}\big)^\beta + \big(\frac{\pi_\theta(y_l|x)}{\pi_{\text{ref}}(y_l|x)}\big)^\beta} \ . \tag{10}$$

The probability assigned to the *loser (rejected)* response $y_l$, denoted by $p_\theta^l$, can be defined similarly. Note that the distribution $p_\theta$ in equation 9 can be thought of as a generalization of the conditional probability of a response $y$ given that $y \in \{y_w, y_l\}$.

**Second**, in the standard classification setting, the dataset gives us access to labels, which we use to extract target probabilities associated with each class. To obtain these target probabilities, we simply use any distribution $p = (p^w, p^l)$ from the simplex $\Delta_2$, which is defined as the set of all vectors $p \in \mathbb{R}^2$ satisfying $p^w, p^l \geq 0$ and $p^w + p^l = 1$. In the most basic case, we just use the one-hot vector $(p^w, p^l) = (+1, 0)$ akin to using *hard labels*. More generally, we can use *soft labels*, meaning that we put some non-zero weight behind each class (Müller et al., 2019).

**Third**, we define a classification loss $\mathcal{L}$ between two distributions $p_\theta$ and $p$, leading us to the optimization problem: $\min_\theta \sum_\mathcal{D} \mathcal{L}(p_\theta, p)$. A good example is the CE loss.

We can now show that a large number of DPO-style algorithms can be viewed as specific instances of this classification framework. The generality arises from the ability to use **(a)** hard or soft labels for the target distribution $p$ and **(b)** different classification losses $\mathcal{L}$.

**Remark 4.1** (DPO). *Suppose that we use the CE loss and hard labels $p := (p^w, p^l) = (+1, 0)$ in the above classification framework. Then, using equation 10, we can write*

$$\mathcal{L}\big(p_\theta, \ p\big) = -\big(p^w \log p_\theta^w + p^l \log p_\theta^l\big) = -\log p_\theta^w = -\log \sigma\left(\beta \log \frac{\pi_\theta(y_w|x)}{\pi_{ref}(y_w|x)} - \beta \log \frac{\pi_\theta(y_l|x)}{\pi_{ref}(y_l|x)}\right) \ ,$$

*which is exactly the DPO loss equation 4 if it is summed over $\mathcal{D}$.*

Another popular DPO-style algorithm is IPO (Azar et al., 2024). While the derivation of IPO in the original paper looks completely different than DPO, we now show that IPO can also be viewed as a specific instance of our classification framework (see Appendix B.1 for the detailed derivation).

**Remark 4.2** (IPO). *We can recover IPO (Eq. 17 in Azar et al. 2024) using the loss $\mathcal{L}\big(p_\theta, \ p\big) = \big(\log(p_\theta^w/p_\theta^l) - \log(p^w/p^l)\big)^2$ and soft labels $p := (p^w, p^l) = (\sigma(1/2), \sigma(-1/2))$.*

Table 1 shows that several DPO-style algorithms can be formulated using this framework. Given this framework, we can formulate the set of optimal solutions (those that achieve 0 loss) for any DPO-style algorithm. An optimal parameter $\theta$ is one that achieves 0 loss for all samples in $\mathcal{D}$, i.e., $p_\theta(x, y_w, y_l) = p, \forall (x, y_w, y_l) \in \mathcal{D}$. Setting $p_\theta^w(x, y_w, y_l)$ in equation 10 equal to $p^w = 1 - \varepsilon$, we obtain

$$\pi_{\theta^*}(y_w|x) = \eta \cdot \pi_{\theta^*}(y_l|x), \quad \text{with} \quad \eta := \sqrt[\beta]{(1-\varepsilon)/\varepsilon} \cdot \frac{\pi_{\text{ref}}(y_w|x)}{\pi_{\text{ref}}(y_l|x)} \ . \tag{11}$$

Note that the derivation using $y_l$ yields the same result. Thus, we have two probabilities, $\pi_{\theta^*}(y_w|x)$ and $\pi_{\theta^*}(y_l|x)$, that we aim to learn, but minimizing the loss only gives us one constraint specified in equation 11. This means that the original classification problem (loss-minimization in DPO-style algorithms) is underspecified. In Figure 2, we provide an illustration of this phenomenon. We also provide a concrete example to further highlight that winner-loser probabilities can move in arbitrary directions, and most notably, can both go to zero.

| Data | Algorithm | Labels | Loss |
|------|-----------|--------|------|
| | `DPO(BT)` | Hard | CE |
| Pairs | `CDPO` | Soft | CE |
| | `IPO` | Soft | Remark 4.2 |
| List (Appendix. B) | `DPO(PL)` | Hard | CE |
| Auxiliary Info | `RPO` | Soft | Appendix B |
| (Appendix. B) | `Distilled DPO` | Soft | Appendix B |

Table 1: Unifying DPO-style algorithms. By `DPO(PL)`, we mean `DPO` with Plackett-Luce model. See our proofs for `RPO` (Adler et al., 2024) and `Distilled DPO` (Fisch et al., 2024) in Appendix B.

**Remark 4.3** (A Concrete Example). *Suppose we have $\varepsilon = 1/11$ and for simplicity we set $\beta = 1$, $\pi_{ref}(y_w|x) = 0.02$, and $\pi_{ref}(y_l|x) = 0.01$, which result in $\eta = 20$. (See Eqn 11 for the definition of $\eta$) We identify two pairs of probabilities that satisfy equation 11:*

1. *$\big(\pi_{\theta^*}(y_w|x),\ \pi_{\theta^*}(y_l|x)\big) = (0.4,\ 0.02)$ .*

2. *$\big(\pi_{\theta^*}(y_w|x),\ \pi_{\theta^*}(y_l|x)\big) = (0.001,\ 0.0002)$ .*

*In the first case both probabilities (including the loser) increase under $\pi_{\theta^*}$ relative to $\pi_{ref}$. In sharp contrast, both probabilities (including the winner) decrease under $\pi_{\theta^*}$ relative to $\pi_{ref}$. Note that the increase is bounded as the two probabilities can go up until they hit $\pi_{\theta^*}(y_w|x) + \pi_{\theta^*}(y_l|x) = (1 + \eta)\pi_{\theta^*}(y_l|x) = 1$. Perhaps more concerning is the observation that the two probabilities can decrease arbitrarily and even collapse to $0$ while still maintaining $\mathcal{L}\big(p_{\theta^*}, p\big) = 0$.*

**List of Preferences**  Our classification framework can can extend to work with lists, rather than pairs, of preferences. In particular, assume that we have $N$ responses for each prompt $x$, giving us a dataset of the form $\mathcal{D} = \{(x, y_1, y_2, \ldots, y_N)\}$. In this case, we can define a list version of the probability vector $p_\theta(x, y_1, \ldots, y_N)$ similarly to equation 9, together with a target distribution $p$. With this simple extension we can now incorporate existing DPO-style algorithms that work with lists. For instance, we can show that `DPO` with the Plackett-Luce model for preferences Rafailov et al. (2023) can be captured in our classification framework, again using hard labels and the CE loss. See Appendix B.3 for a proof.

**Auxiliary Information**  A second important extension pertains to the definition of soft labels in our classification setting. So far we have only worked with soft labels that are fixed across the entire dataset, for instance, $p := (p^w, p^l) = (\sigma(1/2), \sigma(-1/2))$ for all $(x, y_w, y_l)$ in `IPO`. These fixed labels are agnostic about any extra information we may have about our data triplets $(x, y_w, y_l)$. However, in some applications we may have access to some auxiliary scores, $s_w, s_l$, (e.g., ratings) associated with each response, which can then be used to enrich our soft labels. More formally, suppose now that our dataset is comprised of $\mathcal{D} = (x, y_w, s_w, y_l, s_l)$. To obtain the soft labels we can employ, for instance, $p = \text{softmax}(s_w, s_l)$. Combining this with the `IPO` loss, we recover `Distilled DPO` (Eq. 7 in Fisch et al. 2024). Using the same soft labels, but with the CE loss recovers `RPO` (see Sec. 3.3.2 in Adler et al. 2024). We provide more details on both algorithms in Appendix B.4. These natural extensions further demonstrate that our classification framework is fairly general as well as sufficiently flexible to capture a large number of existing DPO-style algorithms.

## 5   Constrained Controlled DPO (`C2-DPO`)

We now present a general family of constraints to control the likelihood displacement of in-sample probabilities in DPO-style algorithms described in Section 3. These constraints can also help with the under-specified nature of these algorithms discussed in Section 4. We define the constraints on the probability mass of the winner-loser pair and use them to control how much this mass changes from the reference policy $\pi_{ref}$ to the target policy $\pi_\theta$. We then show how these constraints can be incorporated into any DPO-style loss function and propose our algorithm, which we refer to as the Constrained Controlled Direct Preference Optimization (`C2-DPO`).

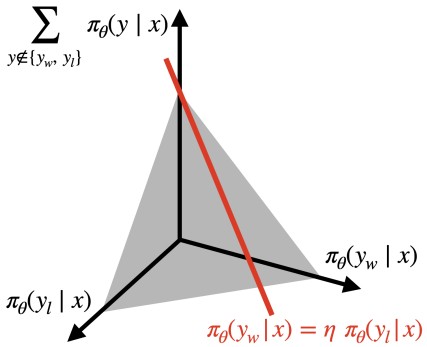

Figure 2: An illustration of the set of solutions that achieve 0 loss in DPO-style algorithms. The shaded gray is the set of feasible solutions. The red line passing through the feasible set indicates the set of optimal solutions. Note that the case where the probability belonging to in-sample responses displace entirely to out-of-sample responses, i.e., $p_\theta(x, y_w, y_l) = (0^+, 0^+)$ also lies on this line.

The constraint, with respect to an arbitrary function $\varphi : \mathbb{R} \to \mathbb{R}$, takes the general form

$$\varphi\big(\pi_\theta(y_w|x)\big) + \varphi\big(\pi_\theta(y_l|x)\big) = \varphi\big(\pi_{\text{ref}}(y_w|x)\big) + \varphi\big(\pi_{\text{ref}}(y_l|x)\big) \;. \tag{12}$$

Note that the RHS is fixed during training, so the two terms on the LHS cannot move in the same direction. We now generalize this intuition by showing that when the constraint function $\varphi$ is monotonic and added to the solution characterization of DPO-style algorithms given by

$$\frac{\pi_{\theta*}(y_w|x)}{\pi_{\text{ref}}(y_w|x)} = \sqrt[\beta]{\frac{1-\varepsilon}{\varepsilon}} \cdot \frac{\pi_{\theta*}(y_l|x)}{\pi_{\text{ref}}(y_l|x)} \;, \tag{13}$$

then we can control the direction of the movement of probability mass for all winner-loser pairs.

**Proposition 5.1.** *Let $\varphi : \mathbb{R} \to \mathbb{R}$ be a monotonic function and assume that equation 12 holds and that the target labels satisfy $(p^w, p^l) = (1-\varepsilon, \varepsilon)$ for some $\varepsilon < 1/2$. Then, $\pi_{\theta*}(y_w|x) > \pi_{ref}(y_w|x)$ and $\pi_{\theta*}(y_l|x) < \pi_{ref}(y_l|x)$.*

Proposition 5.1, whose proof is reported in Appendix C.1, shows that any monotonic constraint function $\varphi$ guarantees that the learned policy, $\pi_{\theta*}$, assigns a higher (lower) probability to the winner (loser) response than the one assigned to it by the reference policy $\pi_{\text{ref}}$. It is natural to ask what $\varphi$ should be used in the context of this constraint, and we present two interesting candidates below.

### 5.1 Logarithmic Constraint $\varphi(x) := \log x$

Our first choice is to employ the logarithmic constraint:

$$\log\left(\pi_\theta(y_w|x)\right) + \log\left(\pi_\theta(y_l|x)\right) = \log\left(\pi_{\text{ref}}(y_w|x)\right) + \log\left(\pi_{\text{ref}}(y_l|x)\right) \;, \tag{14}$$

which is nice to work with empirically in light of the fact that all terms are in the log-space. Moreover, these log probabilities are already computed in `DPO`, which makes the implementation of the corresponding `C2-DPO` algorithm more efficient.

Rather than using hard constraints, we add the deviation from the constraint as a soft penalty to the original DPO-style loss with a regularization parameter $\lambda$. We consider both $\ell_1$ and $\ell_2$ penalties; for conciseness, we present the $\ell_2$ version here. Equipping the `DPO` loss equation 4 with the logarithmic constraint equation 14, we obtain the following loss for `C2-DPO`:

$$\min_\theta \sum_{\mathcal{D}} -\log\sigma\left(\beta\log\frac{\pi_\theta(y_w|x)}{\pi_{\text{ref}}(y_w|x)} - \beta\log\frac{\pi_\theta(y_l|x)}{\pi_{\text{ref}}(y_l|x)}\right) + \lambda\left(\log\frac{\pi_\theta(y_w|x)}{\pi_{\text{ref}}(y_w|x)} + \log\frac{\pi_\theta(y_l|x)}{\pi_{\text{ref}}(y_l|x)}\right)^2. \tag{15}$$

The $\ell_1$ version simply replaces the squared term with an absolute value. We evaluate both variants in Section 6. In contrast to the hard constraint, in this case we do not necessarily force the winner (loser) probability to go up (down). Rather, we impose a penalty when the learner violates the constraint. Notice also that we added the penalty term to the original loss of `DPO` for simplicity, but in principle, the penalty term can be added to any DPO-style loss covered in our classification framework.

Further, we can show that employing the logarithmic constraint equation 14 has a meaningful RLHF interpretation. Recall that Rafailov et al. (2023) defined $\hat{r}_\theta(x, y) := \beta \log \left( \pi_\theta(y|x)/\pi_{\text{ref}}(y|x) \right), \ \forall y \in \mathcal{Y}$ as an *implicit reward* learned during DPO training. Using this notation, we can rewrite objective equation 15 simply as

$$- \log \sigma \left( \hat{r}_\theta(x, y_w) - \hat{r}_\theta(x, y_l) \right) + \frac{\lambda}{\beta^2} \left( \hat{r}_\theta(x, y_w) + \hat{r}_\theta(x, y_l) \right)^2 .$$

Under $\varphi(x) = \log x$, we solve the original RLHF problem akin to `DPO`, but we also incentivize the sum of the implicit rewards for the winner and loser to remain around zero. It follows that the constraint regularizes the implicit rewards so as to avoid rewards that are **(a)** very large and **(b)** have the same sign. These two properties cannot co-exist when employing $\varphi(x) = \log x$, since doing so would yield a large magnitude inside the square and ultimately a large magnitude in the second term of the loss. Intuitively, this can hedge against the likelihood displacement, because in the case of displacement both implicit rewards are large in magnitude and both have a negative sign, which the constraint will greatly penalize.

## 5.2 Identity Constraint $\varphi(x) := x$

A second interesting choice would be to simply use the identity constraint:

$$\pi_\theta(y_w|x) + \pi_\theta(y_l|x) = \pi_{\text{ref}}(y_w|x) + \pi_{\text{ref}}(y_l|x) . \tag{16}$$

While equation 16 is also a plausible constraint, at first glance it is unclear how to implement it since the constraint is no longer in the log-space and is specified in terms of raw probabilities. Working with raw probabilities is prone to numerical underflow issues; thus, we would like to derive a constraint which is equivalent to equation 16 and operates in the log-space. To do so, we make use of the following lemma whose proof is reported in Appendix C.2.

**Lemma 5.1.** *For any two numbers $a$ and $b$, we have $\log(a + b) = \log a - \log \sigma(\log a - \log b)$.*

Applying log to both sides of equation 16, we obtain $\log(\pi_\theta(y_w|x) + \pi_\theta(y_l|x)) = \log(\pi_{\text{ref}}(y_w|x) + \pi_{\text{ref}}(y_l|x))$, which can be rewritten using Lemma 5.1 as

$$\log \left( \pi_\theta(y_w|x) \right) - \log \sigma \left( \log \frac{\pi_\theta(y_w|x)}{\pi_\theta(y_l|x)} \right) = \log \left( \pi_{\text{ref}}(y_w|x) \right) - \log \sigma \left( \log \frac{\pi_{\text{ref}}(y_w|x)}{\pi_{\text{ref}}(y_l|x)} \right) . \tag{17}$$

Moving from equation 16 to equation 17, we have rewritten the constraint entirely in the log-space, thus avoiding numerical issues, and similar to the logarithmic constraint in Section 5.1, allowing a straightforward implementation of the corresponding `C2-DPO` algorithm. Equipping the `DPO` loss equation 4 with the identity constraint equation 17, we obtain the following loss for `C2-DPO` using an $\ell_2$ penalty:

$$\min_\theta \sum_{\mathcal{D}} - \log \sigma \left( \beta \log \frac{\pi_\theta(y_w|x)}{\pi_{\text{ref}}(y_w|x)} - \beta \log \frac{\pi_\theta(y_l|x)}{\pi_{\text{ref}}(y_l|x)} \right)$$
$$+ \lambda \left( \log \frac{\pi_\theta(y_w|x)}{\pi_{\text{ref}}(y_w|x)} + \log \sigma \left( \log \frac{\pi_{\text{ref}}(y_w|x)}{\pi_{\text{ref}}(y_l|x)} \right) - \log \sigma \left( \log \frac{\pi_\theta(y_w|x)}{\pi_\theta(y_l|x)} \right) \right)^2 . \tag{18}$$

An $\ell_1$ variant can be obtained by replacing the squared term with absolute value. We evaluate both penalties in our experiments in Section 6. Note that unlike $\varphi(x) = \log x$, deriving an RLHF interpretation under $\varphi(x) = x$ is subtle. That said, an interesting property under $\varphi(x) = x$ is that the winner probability can increase only by an amount equal to $\pi_{\text{ref}}(y_l|x)$. This means that we will not put the entire probability mass on $y_w$.

## 6 Experiments

In this section, we present results across multiple evaluation settings: head-to-head win rates comparing model outputs (Figure 3), win rates against the preferred responses in the datasets (Figure 4, 6, 8), and holistic evaluation via MT-Bench (Figure 5, 7) to demonstrate consistent improvements across diverse metrics (Zheng et al., 2023). Importantly, `C2-DPO` achieves these improvements with negligible additional computational cost compared to `DPO`, as our constraints only operate on in-sample responses.

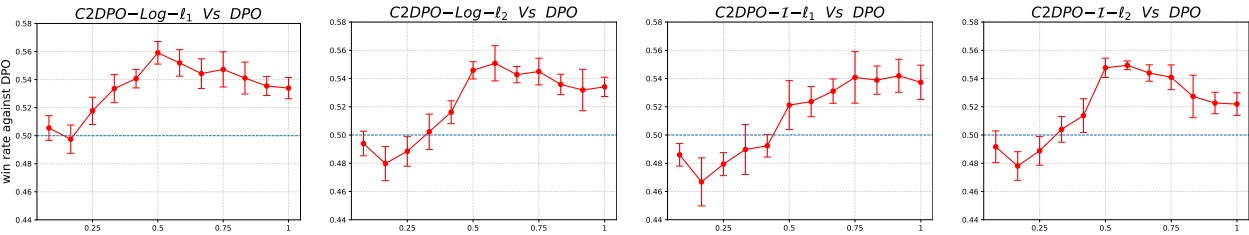

Figure 3: Head-to-head win-rate of `C2-DPO` against `DPO`. For each individual plot, we take the prompts from the held-out test set of Ultrafeedback Binarized and generate responses from the C-3DPO and DPO model. Then we present the prompt and two responses to Anthropic Claude Sonnet and ask which of the two answers is more helpful and honest. We compute the average win rates across 10 different random seeds.

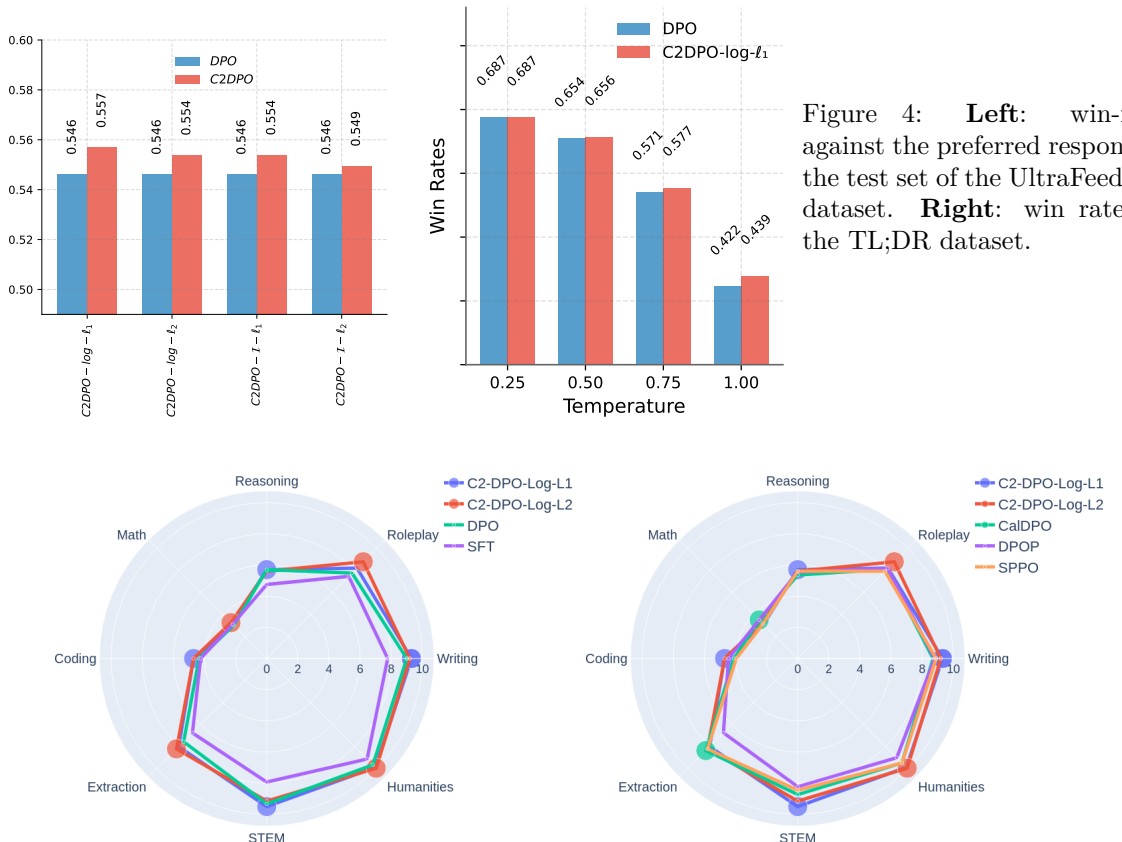

Figure 4: **Left**: win-rates against the preferred response in the test set of the UltraFeedback dataset. **Right**: win rates on the TL;DR dataset.

Figure 5: A comparison between `C2-DPO` and baselines. (Left) The two variants of `C2-DPO-Log` better align Zephyr-7b-SFT relative to vanilla `DPO` as well as the other baselines (Right).

## 6.1 Ultrafeedback Binarized

This dataset is comprised of 64k prompts, and a winner (preferred) and loser (rejected) continuation per prompt ($y_w$ and $y_l$). We used the standard DPO implementation published with the paper (Mitchell, 2023). Following Rasul et al. (2024), we train Zephyr-7B-SFT.

We present the head-to-head win rate of `C2-DPO` against `DPO` in Figure 3 by using 200 prompts from the held-out test set. We compute the win-rate by asking Anthropic's Claude Sonnet-3.5-v2 which response is more helpful. The exact prompt used for Claude is in Appendix E.

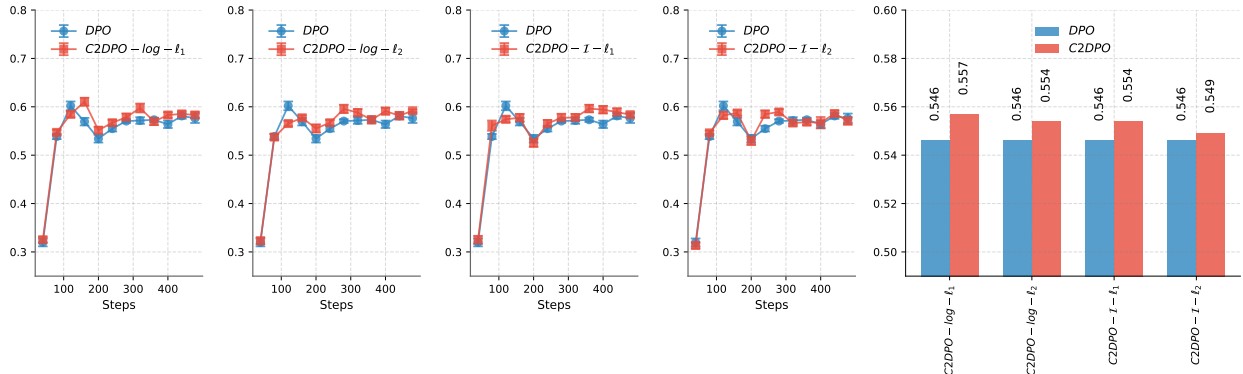

Figure 6: Win rates comparison of Zephyr-7B-SFT aligned on Ultrafeedback Binarized using `DPO` and `C-2DPO`. The first 4 plots show win rates of `C-2DPO` at individual checkpoints across different training trajectories. The last plot shows mean and standard error of win rates across all checkpoints and all inference runs.

Recall from Section 5 that we proposed two candidates for the constraint function $\varphi$, namely the logarithmic and identity functions. Moreover, notice from (18) that we measure the deviation from the constraint using an $\ell_2$ penalty. Alternatively, we can measure this deviation using an $\ell_1$ penalty. Altogether, we have four specific implementations of `C2-DPO`: `C2-DPO-Log-`$\ell_1$, `C2-DPO-Log-`$\ell_2$, `C2-DPO-`$\mathcal{I}$-$\ell_1$, and `C2-DPO-`$\mathcal{I}$-$\ell_2$. In Appendix D, we include pseudo-code for each of these 4 variations. In Figure 3, we show the head to head win rate of each of these 4 implementations against `DPO`, and in Figure 4 (Left), we show win-rate against the preferred response from the dataset. For all 4 implementations, we used the hyper-parameter $\lambda = 2 \times 10^{-4}$ and did not tune it for each of the 4 implementations separately. From this result, it is clear that all 4 variations improve upon DPO, with `C2-DPO-Log` being the highest performer. Figure 6 shows win rates comparison between Zephyr-7B-SFT aligned with `DPO` and `C-2DPO`. We align Zephyr-7B-SFT following Rasul et al. (2024) using `DPO`, `C2-DPO-Log-`$\ell_1$, `C2-DPO-Log-`$\ell_2$, `C2-DPO-`$\mathcal{I}$-$\ell_1$, and `C2-DPO-`$\mathcal{I}$-$\ell_2$ for one epoch, all `C-2DPO` use hyper-parameter $\lambda = 2 \times 10^{-4}$. With each checkpoint, we generate responses using test split of Ultrafeedback Binarized using hyperparameters max_tokens=1000, temperature=1.0, top_p=0.9, top_k=50. Different from the head to head setting, we ask Claude to compare the generated response directly with the preferred response in the dataset. The win rates and standard errors are calculated based on 10 different inference runs.

Further, to holistically evaluate the final model, we used MT-Bench, a multi-turn benchmark that uses GPT-4 to judge models' performance in 8 different categories: Writing, Roleplay, Reasoning, Math, Coding, Extraction, STEM, and Humanities Rasul et al. (2024). In Figure 5 we show the MT-Bench evaluation for `C2-DPO` against vanilla `DPO`, as well as related DPO-style algorithms that discuss the collapse of probabilities in `DPO` and aim to mitigate it, such as `Cal-DPO` (Xiao et al., 2024), `SPPO` (Wu et al., 2024), and `DPOP` (Pal et al., 2024). `C2-DPO` is the most competitive algorithm.

We then used `C2-DPO-Log` on a larger initial checkpoint, namely the Olmo-13B-SFT model from AllenAI OLMo et al. (2024). We compared `C2-DPO-Log` against `DPO`. In Figure 7, we see that `C2-DPO-Log` outperforms `DPO`, indicating that `C2-DPO` improvement may scale to larger models.

Across all evaluation settings on the UltraFeedback dataset, the four `C2-DPO` variants consistently outperform vanilla DPO. The logarithmic constraint variants (`C2-DPO-Log-`$\ell_1$ and `C2-DPO-Log-`$\ell_2$) achieve the strongest performances, with head-to-head win rates around 55% against DPO and improvements across all MT-Bench categories. Notably, all variants use the same hyperparameter without individual tuning, suggesting the method is robust to the choice of constraint function and penalty. The improvements on the larger Olmo-13B model indicate that the benefits of controlling likelihood displacement extend beyond the 7B scale.

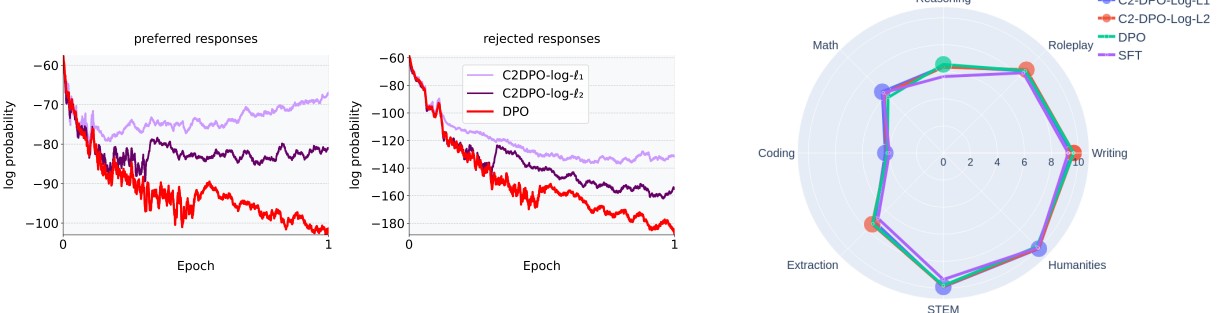

Figure 7: **Left**: A comparison between `C2-DPO` and `DPO` in terms of the probability of winner and loser responses when training the GPT-J model with the Reddit TL;DR dataset. **Right**: Comparison between `C2-DPO` and `DPO` for aligning the Olmo-13B-SFT model using the Ultrafeedback-Binarized dataset.

## 6.2 Reddit TL;DR

We then evaluate our proposed method on a summarization task with human-assigned scores for pairs of summaries. For this purpose, we employ the Reddit TL;DR dataset from Stiennon et al. (2022). We follow Amini et al. (2024) in creating the dataset and include pairs of summaries where one received a higher quality score than the other. During training and testing, we select the highest-scoring summary as $y_w$ and a randomly selected alternative as $y_l$.

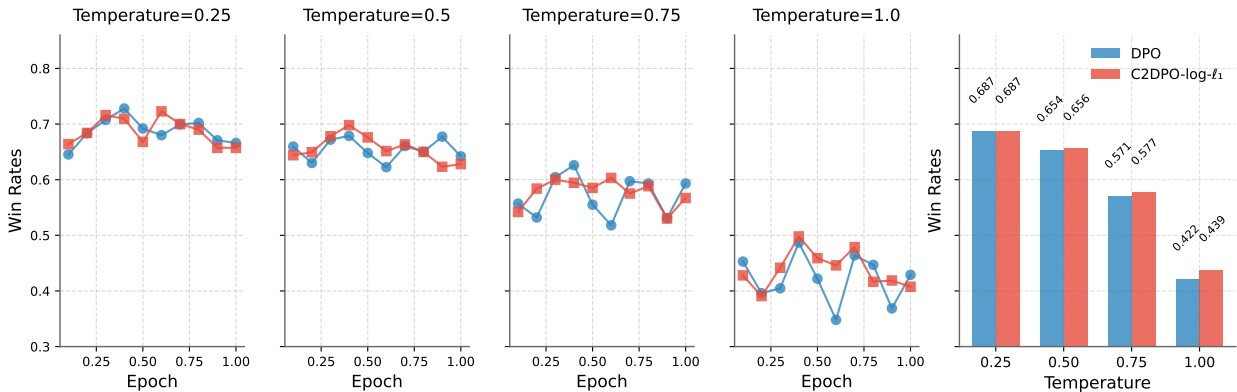

Figure 8: Win rates against $y_w$ comparison of GPT-J aligned on TL;DR using `DPO` and `C-2DPO`. The first 4 plots show win rates of `C-2DPO` at individual checkpoints across different training trajectories. The last plot shows mean and standard error of win rates across all checkpoints and all inference runs.

We align GPT-J Wang & Komatsuzaki (2021) with vanilla `DPO` as well as `C2-DPO` algorithms. Specifically, we first run one SFT epoch with the Reddit TL;DR dataset on GPT-J, and then perform subsequent preference alignment. We evaluate the final checkpoints by computing win rates against $y_w$ following Rafailov et al. (2023); Amini et al. (2024). We use Claude Sonnet 3.5 v2 as a judge, and provide prompt used for Claude in the Appendix F. As shown in Figure 8, `C2-DPO` variants consistently outperform vanilla `DPO` on this summarization task. The improvements demonstrate that controlling likelihood displacement is beneficial beyond conversational tasks, extending to specialized domains like summarization. Figure 7 (Left) further shows that `C2-DPO` successfully mitigates the probability reduction of both preferred and rejected responses during training, consistent with our theoretical predictions from Section 3.

## 7  Related Work

A few recent papers have proposed a unifying perspective on DPO-style algorithms. Notably, Tang et al. (2024) presented a generalization of `DPO` where different supervised learning losses are applied to the difference of implicit rewards $(\hat{r}(x, y_w) - \hat{r}(x, y_l))/\beta$. In contrast, we make a clear connection between DPO-style algorithms and classification, and also extend our results to lists and auxiliary information, as do similar efforts by (Su et al., 2025; Zhao et al., 2025; Im & Li, 2024; Yao et al., 2025). A second notable example was to generalize from KL to any f-divergence when measuring the discrepancy between the target and reference model (Han et al., 2024). We also note that the first ingredient of our classification framework - $\pi_\theta$ - was used by Sharifnassab et al. (2024) to propose a soft version of `DPO`.

Earlier work studied the decrease of likelihood during training (Feng et al., 2024; Xie et al., 2024), explained this phenomenon from different angles, and proposed different losses to address it. Here we provide a brief overview of a number of these results, especially those that we experimentally compare against. Pal et al. (2024) proposed `DPOP` which addresses the phenomenon by adding the penalty term $\max(0, -\hat{r}(x, y_w)/\beta)$ within the log-sigmoid of the `DPO` loss equation 4. `DPOP` can also be viewed as `DPO` with a modified BT model. In this sense, it has similarities with the $\alpha$-`DPO` (Shen et al., 2024) (see also (Choi et al., 2025; Shao et al., 2025)).

Xiao et al. (2024) attributed the undesirable behavior to the contrastive loss of `DPO` not being *scale-calibrated*, i.e., ignoring the absolute values of implicit rewards $\hat{r}(x, y_w)$ and $\hat{r}(x, y_l)$. They address this by constraining the implicit rewards to a scale that matches the ground-truth reward $r$. Thus, in their proposed loss, `Cal-DPO`, they add the square-loss $(\hat{r}(x, y) - r(x, y))^2$, $y \in \{y_w, y_l\}$ to the DPO loss (without $\beta$). Of course, since they do not have access to the ground-truth reward, they replace it with $1/2$ and $-1/2$ for $y_w$ and $y_l$, respectively. A loss similar to `Cal-DPO` was proposed by Wu et al. (2024) and they named it `SPPO`. It is simply `Cal-DPO` without the `DPO` loss. Finally, D'Oosterlinck et al. (2024) proposed `APO`, which offer fine-grained control over the implicit rewards.

While our work focuses on the RLHF setting, some underlying concepts, including pairwise Bradley-Terry modeling and non-identifiability of scores from pairwise comparisons, are well-established in the learning-to-rank and preference learning literature (Newman, 2023; Hamilton et al., 2025; Casper et al., 2023; Zhu et al., 2024; Khorasani et al., 2025). A key distinction is that in RLHF, scores are not free parameters but implicit rewards tied to a policy through $r(x, y) = \beta \log(\pi_\theta(y|x)/\pi_{\mathrm{ref}}(y|x))$. This coupling means that manipulating probability ratios directly affects the learned policy, unlike standard ranking formulations. Our contributions, including proving that DPO derives from a subset-constrained KL objective (Section 3), establishing the resulting likelihood displacement with an explicit $e^{-1}$ bound (Lemma 3.2), and proposing constraints that exploit the $(\pi_\theta, \pi_{\mathrm{ref}})$ structure to control this displacement (Section 5), are specific to policy optimization with preference data in RLHF.

## 8  Conclusion & Future Work

In this work, we revisited the derivation of DPO and revealed two counter-intuitive findings. We first proved that the standard DPO loss can arise from a formulation that regularizes only in-sample responses. This, in turn, uncovered another surprising result: both the preferred and rejected responses are likely to experience a decrease in likelihood. This insight shed light on the likelihood-displacement phenomenon observed in prior empirical studies. We then generalized this result through a unifying classification framework, highlighting a broader absence of out-of-sample KL regularization. Building on these findings, we introduced the Constrained Controlled DPO (`C2-DPO`), a principled extension of `DPO` that controls probability displacement by constraining the redistribution of likelihood between preferred and rejected responses. `C2-DPO` retains the computational efficiency of standard `DPO` while still offering a clear RLHF interpretation.

We know that `DPO` displaces probabilities to unseen samples, but we do not have a clear picture in terms of the kinds of responses for which the probability increases. While Fisch et al. (2024) and Razin et al. (2025) present some findings for special cases, it would be interesting to do a more systematic study of the kinds of unseen responses whose likelihood increases during training.

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

## A  Proof of Lemmas 3.1 and 3.2

In order to prove Lemma 3.1 we will need first the following technical result. We recall that $\Delta_n$ denotes the simplex in $\mathbb{R}^n$ meaning the set of all vectors $p \in \mathbb{R}^n$ for which $\sum_{i=1}^n p_i = 1$ and $p \geq 0$.

**Lemma A.1.** *Let $q \in \Delta_n$ and $r \in \mathbb{R}^n$. Let $S \subseteq [n] := \{1, 2, \dots, n\}$ be a subset for which $|S| \geq 2$. Any optimal solution $p$ of the following problem*

$$\arg\max_{p \in \Delta_n} \left\{ r^T p - \beta \sum_{i \in S} p_i \ln \frac{p_i}{q_i} \right\} . \tag{19}$$

*satisfies that $\beta \left( \ln (p_i/q_i) - \ln (p_j/q_j) \right) = r_i - r_j$ holds true for any $i, j \in S$.*

*Proof.* The problem equation 19 is convex and Slater's condition is satisfied; hence, the set of optimal solutions coincides with the set of KKT points. Therefore, we will consider the Lagrangian

$$L(p, \lambda, \mu) = r^T p - \beta \sum_{i \in S} p_i \ln \frac{p_i}{q_i} + \lambda^T p + \mu \left( \sum_{i=1}^n p_i - 1 \right) , \tag{20}$$

where $\lambda \in \mathbb{R}_+^n$ and $\mu \in \mathbb{R}$ are the Lagrange multipliers. The KKT conditions are

$$\begin{cases} \frac{\partial L}{\partial p_i} = r_i - \beta \left( \ln \frac{p_i}{q_i} + 1 \right) + \lambda_i + \mu = 0, & \forall i \in S \\ \frac{\partial L}{\partial p_i} = r_i + \lambda_i + \mu = 0, & \forall i \notin S, \\ p_i \geq 0, \quad \sum_{i=1}^n p_i = 1, & \text{(feasibility)}, \\ \lambda_i \geq 0, & \forall i \in [n], \quad \text{(multipliers)}, \\ \lambda_i p_i = 0, & \forall i \in [n], \quad \text{(complementary slackness)}. \end{cases} \tag{21}$$

Let $i \in S$, after rearranging the condition we get that $p_i = q_i e^{(r_i + \lambda_i + \mu)/\beta - 1}$. Moreover, if $q_i > 0$ then $p_i > 0$ and hence $\lambda_i = 0$. On the other hand, if $q_i = 0$ then $p_i = 0$ and the value of $\lambda_i$ does not matter. Hence, for simplicity, we can take $\lambda_i = 0$ for all $i \in S$. Thus, the KKT conditions can be equivalently written as follows

$$\begin{cases} p_i = q_i e^{(r_i + \mu)/\beta - 1}, & \forall i \in S. \\ r_i + \lambda_i + \mu = 0, & \forall i \notin S, \\ p \in \Delta_n, & \text{(feasibility)}, \\ \lambda_i \geq 0, & \forall i \notin S, \quad \text{(multipliers)}, \\ \lambda_i p_i = 0, & \forall i \notin S, \quad \text{(complementary slackness)}. \end{cases} \tag{22}$$

Let $\hat{r} = \max_{i \notin S} r_i$. We now split the proof into the two cases:

- Case I: $\hat{r} > \beta \ln \sum_{i \in S} q_i e^{r_i/\beta - 1}$

  First, for any $i \notin S$, since $\lambda_i \geq 0$ we have that $\mu \leq -r_i$ and therefore in particular $\mu \leq -\hat{r}$. Thus, there must be $i \notin S$ for which $p_i > 0$. Indeed, if this is not the case then $p_i = 0$ for all $i \notin S$ and we get a contradiction since (recall that $\mu \leq -\hat{r}$)

  $$1 = \sum_{i=1}^n p_i = \sum_{i \in S} p_i = \sum_{i \in S} q_i e^{(r_i + \mu)/\beta - 1} \leq \sum_{i \in S} q_i e^{(r_i - \hat{r})/\beta - 1} < 1 , \tag{23}$$

  where the last inequality follows from the condition of Case I. Therefore, for simplicity we take $i_* \notin S$ for which $p_{i_*} > 0$. Therefore, $\lambda_{i_*} = 0$.

  Now, for all $j \notin S$, we have $r_{i_*} + \lambda_{i_*} = r_j + \lambda_j$ and thus $r_{i_*} = r_j + \lambda_j \geq r_j$, which means that $r_{i_*} = \hat{r}$. Hence $\mu = -\hat{r}$, and therefore $p_i = q_i e^{(r_i - \hat{r})/\beta - 1}$ for all $i \in S$.

Moreover, this shows that if for some $i \notin S$ we have $r_i < \hat{r}$, then $p_i = 0$. Therefore, by using the feasibility condition we obtain that any KKT point can be described by

$$p_i = q_i e^{(r_i - \hat{r})/\beta - 1}, \quad \forall i \in S$$
$$p_i = 0, \quad \forall i \notin S, r_i < \hat{r}$$
$$\sum_{j \notin S, r_j = \hat{r}} p_j = 1 - \sum_{j \in S} q_j e^{(r_j - \hat{r})/\beta - 1} .$$

- Case II: $\hat{r} \leq \beta \ln \sum_{i \in S} q_i e^{r_i/\beta - 1}$

  In this case, we will first prove that $p_i = 0$ for all $i \notin S$. Suppose in contradiction that $p_\ell > 0$ for some $\ell \notin S$. Then, $\lambda_\ell = 0$ and hence $\mu = -r_\ell$. Recall that $r_\ell \leq \hat{r}$, we obtain that

$$1 = \sum_{i=1}^{n} p_i \geq p_\ell + \sum_{i \in S} p_i = p_\ell + \sum_{i \in S} q_i e^{(r_i - r_\ell)/\beta - 1} \geq p_\ell + \sum_{i \in S} q_i e^{(r_i - \hat{r})/\beta - 1} \geq p_\ell + 1 > 1 ,$$

  where the third inequality follows from the condition of Case II. Therefore, the unique KKT point is given by

$$p_i = \frac{q_i e^{r_i/\beta}}{\sum_{j \in S} q_j e^{r_j/\beta}}, \quad \forall i \in S$$
$$p_i = 0, \quad \forall i \notin S .$$

Finally, we see that in both cases, we have, for any $i, j \in S$, that

$$\beta \left( \ln \left( p_i/q_i \right) - \ln \left( p_j/q_j \right) \right) = r_i - r_j,$$

as required. $\qquad \square$

Now, we can prove Lemma 3.1. To this end, we recall the new optimization problem that we study

$$\max_{\theta} \ \mathbb{E}_x \left[ \mathbb{E}_{y \sim \pi_\theta} \left[ r_\phi(x, y) \right] - \beta \sum_{y \in S_x} \pi_\theta(y|x) \log \frac{\pi_\theta(y|x)}{\pi_{\mathrm{ref}}(y|x)} \right] . \tag{24}$$

*Proof of Lemma 3.1.* First, note that the optimization problem equation 5 is separable in $x$, so we can fix some prompt $x$ and consider the problem

$$\max_{\pi_\theta} \ \mathbb{E}_{y \sim \pi_\theta(x)} \left[ r_\phi(x, y) \right] - \beta \sum_{y \in S_x} \pi_\theta(y|x) \tag{25}$$

From Lemma A.1, the optimization problem equation 24 admits a closed-form solution , which depends on the relation between $\hat{r}_x := \max_{y \notin S_x} r_\phi(x, y)$ and the rewards within $S_x$. In the case where

$$\hat{r}_x > \beta \ln \left( \sum_{y \in S_x} \pi_{\mathrm{ref}} \left( y \mid x \right) e^{\frac{1}{\beta} r_\phi(x, y) - 1} \right) , \tag{26}$$

a policy $\pi_\theta$ is an optimal solution to equation 25 if and only if

$$\pi_\theta \left( y \mid x \right) = \pi_{\mathrm{ref}} \left( y \mid x \right) \exp \left( \frac{1}{\beta} \left( r_\phi \left( x, y \right) - \hat{r}_x \right) - 1 \right), \quad \forall y \in S_x \tag{27}$$

$$\sum_{y \notin S_x, r_\phi(x, y) = \hat{r}_x} \pi_\theta \left( y \mid x \right) = 1 - \sum_{y \in S_x} \pi_{\mathrm{ref}} \left( y \mid x \right) \exp \left( \frac{1}{\beta} \left( r_\phi \left( x, y \right) - \hat{r}_x \right) - 1 \right) \tag{28}$$

$$\pi_\theta \left( y \mid x \right) = 0, \quad \forall y \notin S_x, r_\phi \left( x, y \right) < \hat{r}_x . \tag{29}$$

In the case where condition equation 26 does not hold, the unique optimal solution to equation 25 is

$$\pi_\theta (y \mid x) = \frac{\pi_{\text{ref}} (y \mid x) e^{\frac{1}{\beta} r_\phi(x,y)}}{\sum_{y' \in S_x} \pi_{\text{ref}} (y' \mid x) e^{\frac{1}{\beta} r_\phi(x,y')}}, \quad \forall y \in S_x \tag{30}$$

$$\pi_\theta (y \mid x) = 0, \quad \forall y \notin S_x . \tag{31}$$

We check that in both cases, for any two responses $y, y' \in S_x$, the optimal policy satisfies:

$$\frac{\pi_\theta(y|x)}{\pi_\theta(y'|x)} = \frac{\pi_{\text{ref}}(y|x)e^{\frac{1}{\beta} r_\phi \text{b} x,y}}{\pi_{\text{ref}}\text{b} y' \mid x e^{\frac{1}{\beta} r_\phi \text{b} x,y'}} \tag{32}$$

Taking logarithms and rearranging:

$$r_\phi(x,y) - r_\phi(x,y') = \beta \ln \frac{\pi_\theta(y|x)}{\pi_{\text{ref}}(y|x)} - \beta \ln \frac{\pi_\theta(y'|x)}{\pi_{\text{ref}}(y'|x)} \tag{33}$$

For any preference triplet $(x, y_w, y_l) \in \mathcal{D}$, applying this with $y = y_w$ and $y' = y_l$ and substituting into the Bradley-Terry model $p(y_w \succ y_l|x) = \sigma(r_\phi(x, y_w) - r_\phi(x, y_l))$ gives:

$$p(y_w \succ y_l|x) = \sigma \left( \beta \ln \frac{\pi_\theta(y_w|x)}{\pi_{\text{ref}}(y_w|x)} - \beta \ln \frac{\pi_\theta(y_l|x)}{\pi_{\text{ref}}(y_l|x)} \right) \tag{34}$$

which are exactly the probabilities used in the `DPO` derivation, and taking the negative-log-likelihood of equation 34 over the entire dataset yields exactly the `DPO` loss. □

We complete this section with the proof of Lemma 3.2.

*Proof of Lemma 3.2.* If $r_\phi(x,y) \leq \hat{r}_x$ for every $y \in S_x$, then the condition $\hat{r}_x > \beta \ln(\sum_{y \in S_x} \pi_{\text{ref}}(y|x)e^{r_\phi(x,y)/\beta-1})$ is satisfied. This is because

$$
\begin{aligned}
\beta \ln \sum_{y \in S_x} \pi_{\text{ref}}(y|x)e^{r_\phi(x,y)/\beta-1} &= \beta \ln \sum_{y \in S_x} \pi_{\text{ref}}(y|x)e^{\hat{r}_x/\beta+(r_\phi(x,y)-\hat{r}_x)/\beta-1} \\
&= \hat{r}_x + \beta \ln \sum_{y \in S_x} \pi_{\text{ref}}(y|x)e^{(r_\phi(x,y)-\hat{r}_x)/\beta-1} \\
&\leq \hat{r}_x + \beta \ln \sum_{y \in S_x} \pi_{\text{ref}}(y|x)e^{-1} \\
&= \hat{r}_x + \beta \left( -1 + \ln \sum_{y \in S_x} \pi_{\text{ref}}(y|x) \right) \\
&\leq \hat{r}_x - \beta ,
\end{aligned}
$$

where the last inequality follows from the fact that $\sum_{y \in S_x} \pi_{\text{ref}}(y|x) \leq 1$. This proves that the condition holds true.

Therefore, as we saw in the proof of Lemma 3.1, in any optimal solution $\pi_\theta$ we have for all $y \in S_x$, that

$$\pi_\theta(y|x) = \pi_{\text{ref}}(y|x)e^{(r_\phi(x,y)-\hat{r}_x)/\beta-1} \leq \pi_{\text{ref}}(y|x)e^{-1} = e^{-1}\pi_{\text{ref}}(y|x),$$

which proves the desired result. □

# B Derivations of Existing Algorithms via the Classification Framework

## B.1 IPO

We recall that `IPO` can be formulated as a classification problem with the soft labels $p := (p^w, p^l) = (\sigma(1/2), \sigma(-1/2))$ and the loss given by

$$\mathcal{L}(p_\theta, \ p) = \left( \log \frac{p_\theta^w}{p_\theta^l} - \log \frac{p^w}{p^l} \right)^2 . \tag{35}$$

To show that we indeed recover the `IPO`, we first note that

$$\frac{p^w}{p^l} = \frac{\sigma(1/2)}{\sigma(-1/2)} = \frac{1 + \exp(1/2)}{1 + \exp(-1/2)} = \exp(1/2) \ ,$$

where the second equality follows from the definition of the sigmoid $\sigma(x) = 1/(1 + \exp(-x))$. Moreover, using the definitions of $p_\theta^w$ (see equation 10) and $p_\theta^l$, we obtain that

$$\frac{p_\theta^w}{p_\theta^l} = \frac{\left(\frac{\pi_\theta(y_w|x)}{\pi_{\text{ref}}(y_w|x)}\right)^\beta}{\left(\frac{\pi_\theta(y_l|x)}{\pi_{\text{ref}}(y_l|x)}\right)^\beta} = \left(\frac{\pi_\theta(y_w|x)\pi_{\text{ref}}(y_l|x)}{\pi_\theta(y_l|x)\pi_{\text{ref}}(y_w|x)}\right)^\beta \ . \tag{36}$$

Plugging these two developments to the loss in equation 35 yields

$$\mathcal{L}(p_\theta, \ p) = \left(\log \frac{p_\theta^w}{p_\theta^l} - \log \frac{p^w}{p^l}\right)^2 = \left(\beta \log \frac{\pi_\theta(y_w|x)\pi_{\text{ref}}(y_l|x)}{\pi_\theta(y_l|x)\pi_{\text{ref}}(y_w|x)} - \frac{1}{2}\right)^2 = \left(\log \frac{\pi_\theta(y_w|x)\pi_{\text{ref}}(y_l|x)}{\pi_\theta(y_l|x)\pi_{\text{ref}}(y_w|x)} - \frac{1}{2\beta}\right)^2 \ ,$$

which is exactly `IPO` (see Eq. (17) of Azar et al. 2024).

## B.2 `CDPO`

Recall the `CDPO` Mitchell (2024) loss is given for any triplet $(x, y_w, y_l)$ by

$$-(1 - \varepsilon) \log \sigma \left(\beta \log \frac{\pi_\theta(y_w|x)}{\pi_{\text{ref}}(y_w|x)} - \beta \log \frac{\pi_\theta(y_l|x)}{\pi_{\text{ref}}(y_l|x)}\right) - \varepsilon \log \sigma \left(\beta \log \frac{\pi_\theta(y_l|x)}{\pi_{\text{ref}}(y_l|x)} - \beta \log \frac{\pi_\theta(y_w|x)}{\pi_{\text{ref}}(y_w|x)}\right) \ ,$$

and then summed over all the triplets in the dataset $\mathcal{D}$. In order to see `CDPO` as a classification with the soft labels $p := (p^w, p^l) = (1 - \varepsilon, \varepsilon)$ and the CE loss, we will use the following technical fact

$$\sigma(\beta \log(a/b)) = \frac{1}{1 + \exp(-\beta \log(a/b))} = \frac{1}{1 + \frac{\exp \log b^\beta}{\exp \log a^\beta}} = \frac{1}{1 + \frac{b^\beta}{a^\beta}} = \frac{a^\beta}{a^\beta + b^\beta}.$$

Therefore, with $a = \frac{\pi_\theta(y_w|x)}{\pi_{\text{ref}}(y_w|x)}$ and $b = \frac{\pi_\theta(y_l|x)}{\pi_{\text{ref}}(y_l|x)}$ we get from equation 10 that $\sigma(\beta \log a/b) = p_\theta^w$. Similarly, we get that $\sigma(\beta \log(b/a)) = p_\theta^l$. Therefore, the `CDPO` loss can be written as follows

$$-(1 - \varepsilon) \log p_\theta^w - \varepsilon \log p_\theta^l = -p^w \log p_\theta^w - p^l \log p_\theta^l \ ,$$

which is exactly the CE loss on the vectors $p_\theta = (p_\theta^w, p_\theta^l)$ and $p = (p^w, p^l)$.

## B.3 `DPO (PL)`

In this setting, we are given a dataset of the form $\mathcal{D} = \{(x, y_1, y_2, \ldots, y_N)\}$. In order to recover, the `DPO` with Plackett-Luce Rafailov et al. (2023), we need to generalize the definition of the probability vector $p_\theta$ from pairs as in equation 10 to the following $N - 1$ subsets of the list, namely the first $N$, then the first $N - 1$, then first $N - 2$ and so on. More precisely, for any $1 \le n < N$ we define

$$p_\theta(x, y_n, y_{n+1}, \ldots, y_N) = \text{softmax}\big((r_\theta(x, y_n), r_\theta(x, y_{n+1}), \ldots, r_\theta(x, y_N))\big) \ .$$

In this case, the hard label vectors are defined, for any $1 \le n < N$, by $p^{[n,N]} := (1, 0, \ldots, 0) \in \mathbb{R}^{N-n+1}$. Now, using the CE loss we get the desired result as follows

$$
\begin{aligned}
-\sum_{n=1}^{N-1} \sum_{i=n}^{N} p_i^{[n,N]} \log p_\theta^i(x, y_n, y_{n+1}, \ldots, y_N) \ &= \ -\sum_{n=1}^{N-1} \log \frac{\exp\left(\beta \log \frac{Z(x)\pi_\theta(y_n|x)}{\pi_{\text{ref}}(y_n|x)}\right)}{\sum_{i=n}^{N} \exp\left(\beta \log \frac{Z(x)\pi_\theta(y_i|x)}{\pi_{\text{ref}}(y_i|x)}\right)} \\
&= \ -\sum_{n=1}^{N-1} \log \frac{\exp\left(\beta \log \frac{\pi_\theta(y_n|x)}{\pi_{\text{ref}}(y_n|x)}\right)}{\sum_{i=n}^{N} \exp\left(\beta \log \frac{\pi_\theta(y_i|x)}{\pi_{\text{ref}}(y_i|x)}\right)} \\
&= \ -\log \prod_{n=1}^{N-1} \frac{\exp\left(\beta \log \frac{\pi_\theta(y_n|x)}{\pi_{\text{ref}}(y_n|x)}\right)}{\sum_{i=n}^{N} \exp\left(\beta \log \frac{\pi_\theta(y_i|x)}{\pi_{\text{ref}}(y_i|x)}\right)} \ .
\end{aligned}
$$

### B.4 `RPO` and `Distilled DPO`

As we discussed in Section 3.2, `RPO` can be reformulated as a classification with soft labels, which are defined by $p = \text{softmax}(s_w, s_l)$. Therefore, we immediately see that

$$p^w = \frac{\exp(s_w)}{\exp(s_w) + \exp(s_l)} = \frac{1}{1 + \exp(-(s_w - s_l))} \ ,$$

and thus $p^w = \sigma(s_w - s_l)$. Similarly, we get that $p^l = \sigma(-(s_w - s_l))$. Thus, `RPO` can be seen as a generalization of `CDPO` where the soft labels are given by a certain score and not fixed. To recover the `RPO` loss we denote $a = \beta \log \frac{\pi_\theta(y_w|x)}{\pi_{\text{ref}}(y_w|x)} - \beta \log \frac{\pi_\theta(y_l|x)}{\pi_{\text{ref}}(y_l|x)}$ and $b = s_w - s_l$. Then, we get that

$$\sigma(b) \log \frac{\sigma(b)}{\sigma(a)} + (1 - \sigma(b)) \log \frac{1 - \sigma(b)}{1 - \sigma(a)} = p^w \log \frac{p^w}{p_\theta^w} + p^l \log \frac{p^l}{p_\theta^l} \ .$$

By eliminating the constant terms (with respect to $\theta$) $p^w \log p^w + p^l \log p^l$, we indeed get the CE loss.

To recover the `Distilled DPO` (Equation (7) of Fisch et al. (2024)), we consider the soft labels $p = \text{softmax}(s_w, s_l)$ with the loss of IPO.

$$\mathcal{L}(p_\theta, \ p) = \left( \log \frac{p_\theta^w}{p_\theta^l} - \log \frac{p^w}{p^l} \right)^2 \ .$$

Indeed, it this case we have that

$$\frac{p^w}{p^l} = \frac{\exp(s_w)}{\exp(s_w) + \exp(s_l)} \cdot \frac{\exp(s_w) + \exp(s_l)}{\exp(s_l)} = \frac{\exp(s_w)}{\exp(s_l)} \ ,$$

and therefore $\log(p^w/p^l) = s_w - s_l$. Combining this with equation 36 yields the desired result.

## C Proofs of Section 5

### C.1 Proof of Proposition 5.1

**Proposition 5.1.** *Let $\varphi : \mathbb{R} \to \mathbb{R}$ be a monotonic function and assume that equation 12 holds and that the target labels satisfy $(p^w, p^l) = (1 - \varepsilon, \varepsilon)$ for some $\varepsilon < 1/2$. Then, $\pi_{\theta^*}(y_w|x) > \pi_{ref}(y_w|x)$ and $\pi_{\theta^*}(y_l|x) < \pi_{ref}(y_l|x)$.*

*Proof.* For simplicity we assume that $\varphi$ is monotonically increasing (the same arguments can be easily applied to the monotonically decreasing case). Since $\varepsilon < 1/2$, we have that $(1 - \varepsilon)/\varepsilon > 1$, and therefore from equation 13, we get that

$$\pi_{\theta^*}(y_w|x) > \frac{\pi_{\theta^*}(y_l|x)}{\pi_{\text{ref}}(y_l|x)} \cdot \pi_{\text{ref}}(y_w|x) \ . \tag{37}$$

We will show that $\pi_{\theta^*}(y_l|x) < \pi_{\text{ref}}(y_l|x)$ by contradiction. To this end, we assume that $\pi_{\theta^*}(y_l|x) \geq \pi_{\text{ref}}(y_l|x)$. From equation 37, we immediately obtain that $\pi_{\theta^*}(y_w|x) > \pi_{\text{ref}}(y_w|x)$. Applying the constraint function $\varphi$ to the following two inequalities:

$$\pi_{\theta^*}(y_w|x) > \pi_{\text{ref}}(y_w|x) \quad \text{and} \quad \pi_{\theta^*}(y_l|x) \geq \pi_{\text{ref}}(y_l|x) \ ,$$

and using its monotonicity, we obtain

$$\varphi\big(\pi_{\theta^*}(y_w|x)\big) > \varphi\big(\pi_{\text{ref}}(y_w|x)\big) \quad \text{and} \quad \varphi\big(\pi_{\theta^*}(y_l|x)\big) \geq \varphi\big(\pi_{\text{ref}}(y_l|x)\big).$$

By adding both sides of the above two inequalities, we get a contradiction to equation 12. Thus, proving that $\pi_{\theta^*}(y_l|x) < \pi_{\text{ref}}(y_l|x)$. Similarly, we can prove that $\pi_{\theta^*}(y_w|x) > \pi_{\text{ref}}(y_w|x)$. □

## C.2 Proof of Lemma 5.1

**Lemma 5.1.** *For any two numbers $a$ and $b$, we have $\log(a + b) = \log a - \log \sigma(\log a - \log b)$.*

*Proof.* First, we write

$$a + b = a \left( \frac{a}{a + b} \right)^{-1} ,$$

which thanks to classical logarithmic rules yields that

$$\log(a + b) = \log a - \log \frac{a}{a + b} = \log a - \log \frac{1}{1 + b/a} .$$

Using the definition of the sigmoid function we get

$$\frac{1}{1 + b/a} = \frac{1}{1 + \exp(\log b - \log a)} = \sigma(\log a - \log b) ,$$

which proves the desired result. □

## D C2-DPO Implementation Details

We show implementation details of C2-DPO-Log-$\ell_1$, C2-DPO-Log-$\ell_2$, C2-DPO-$\mathcal{I}$-$\ell_1$, and C2-DPO-$\mathcal{I}$-$\ell_2$ below.

```python
def dpo_loss(pi_logps, ref_logps, yw_idxs, yl_idxs, beta, algo_name, reg_coeff):
    """
    pi_logps: policy logprobs, shape (B,)
    ref_logps: reference model logprobs, shape (B,)
    yw_idxs: preferred completion indices in [0, B-1], shape (T,)
    yl_idxs: dispreferred completion indices in [0, B-1], shape (T,)
    beta: temperature controlling strength of KL penalty
    Each pair of (yw_idxs[i], yl_idxs[i]) represents the indices of a single
    preference pair.
    """
    pi_yw_logps,  pi_yl_logps =  pi_logps[yw_idxs],  pi_logps[yl_idxs]
    ref_yw_logps, ref_yl_logps = ref_logps[yw_idxs], ref_logps[yl_idxs]
    pi_logratios  = pi_yw_logps - pi_yl_logps
    ref_logratios = ref_yw_logps - ref_yl_logps
    losses = -F.logsigmoid(beta * (pi_logratios - ref_logratios))
    if algo_name == 'c2dpo_log_l1':
        reguralized_losses_without_square = \
            pi_yw_logps + pi_yl_logps - ref_yw_logps - ref_yl_logps
        reguralized_losses = (reguralized_losses_without_square) ** 2
        losses = losses + reg_coeff * reguralized_losses
    elif algo_name == 'c2dpo_log_l2':
        reguralized_losses = F.l1_loss(
            pi_yw_logps + pi_yl_logps,
            ref_yw_logps + ref_yl_logps
        )
        losses = losses + reg_coeff * reguralized_losses
    elif algo_name == 'c2dpo_i_l1':
        reguralized_losses_without_square = \
            pi_yw_logps - F.logsigmoid(pi_yw_logps - pi_yl_logps) - \
                (ref_yw_logps - F.logsigmoid(ref_yw_logps - ref_yl_logps))
        reguralized_losses = (reguralized_losses_without_square) ** 2
        losses = losses + reg_coeff * reguralized_losses
```

```
elif algo_name == 'c2dpo_i_l2':
    reguralized_losses = F.l1_loss(
        pi_yw_logps - F.logsigmoid(pi_yw_logps - pi_yl_logps),
        ref_yw_logps - F.logsigmoid(ref_yw_logps - ref_yl_logps)
    )
    losses = losses + reg_coeff * reguralized_losses
rewards = beta * (pi_logps - ref_logps).detach()
return losses, rewards
```

## E  Ultrafeedback Binarized **Claude 3.5 Sonnet v2 win rate prompt and hyperparameters**

In this section we include the prompt used to generate win rates for the Ultrafeedback Binarized experiments. We use Claude Sonnet 3.5 v2 (AWS Bedrock model ID anthropic.claude-3-5-sonnet-20241022-v2:0) to generate win rates. We set the max_tokens to 1024, temperature to 0, and used default value 0.999 for top_p and top_k disabled.

```
For the following query to a chatbot, which response is more helpful?

Query: <prompt>

Response A:
<one of the responses>

Response B:
<the other response>

FIRST provide a one-sentence comparison of the two responses and explain
which you feel is more helpful. SECOND, on a new line, state only "A" or "B"
to indicate which response is more helpful. Your response should use the format:
Comparison: <one-sentence comparison and explanation>
More helpful: <"A" or "B">
```

## F  Reddit TL;DR **Claude 3.5 Sonnet v2 win rate prompt and hyperparameters**

In this section we include the prompt used to generate win rates for the Reddit TL;DR experiments. We use Claude Sonnet 3.5 v2 (AWS Bedrock model ID anthropic.claude-3-5-sonnet-20241022-v2:0) to generate win rates. We set the max_tokens to 1024, temperature to 0, and used default value 0.999 for top_p and top_k disabled.

```
Which of the following summaries does a better job of summarizing the most
important points in the given forum post, without including unimportant or
irrelevant details? A good summary is both precise and concise.
Post: {prompt}
Summary A:
{baseline_response}
Summary B:
{generated_response}
FIRST provide a one-sentence comparison of the two summaries, explaining
which you prefer and why. SECOND, on a new line, state only "A" or "B" to
indicate your choice. Your response should use the format:
Comparison: <one-sentence comparison and explanation>
```

```
Preferred: <"A" or "B">
```

