# OpenReview forum: "$$\texttt{C2-DPO}$$: Constrained Controlled Direct Preference Optimization"
_TMLR — Accepted by TMLR_

### Review · Reviewer_vUaj · 2025-11-17

**Summary Of Contributions:**

This paper provides two counter-intuitive observations about Direct Preference Optimization (DPO): first, the DPO loss can be derived from an alternative optimization problem that only applies a KL guardrail to in-sample responses. Second, this alternative formulation theoretically explains the observed phenomenon where both preferred and rejected responses tend to decrease in probability (likelihood displacement) during DPO training. Based on these insights, the authors propose Constrained Controlled DPO (C2-DPO), which introduces constraints to limit this probability displacement. C2-DPO is shown to outperform vanilla DPO and several baselines on standard alignment datasets.

**Audience:**

Yes

**Audience Explanation:**

The alignment problem and RLHF/DPO algorithms are central topics in current ML research. The paper's theoretical dissection of DPO's subtle mechanism and its counter-intuitive side effect (likelihood displacement) is of academic interest. Furthermore, proposing C2-DPO as a simple, effective, and computationally efficient fix addresses a practical limitation of a widely used algorithm, which may be appealing to practitioners.

**Claims And Evidence:**

Yes

**Claims Explanation:**

Mostly yes. The theoretical derivations (Lemma 3.1, 3.2, Proposition 5.1) are key and seem sound, especially the proof connecting DPO loss to the in-sample-only KL regularization. The empirical evidence shows consistent improvements of C2-DPO variants over vanilla DPO in win rates and holistic MT-Bench scores. Figures clearly support the core claim about in-sample probability reduction in DPO and mitigation in C2-DPO. The classification framework for unifying DPO-style algorithms is well-supported in the appendix.

**Requested Changes:**

I generally feel this paper is well-executed. Please correct me if I'm wrong about anything I've said.

1. The paper introduces $\ell_1$ and $\ell_2$ penalties for the regularization term in Section 5.1, but the loss equations (12 and 15) only show the squared penalty. Please clarify this inconsistency in the main text, or ensure that the generic $\ell_1$ version is also formally presented. (The implementation details in Appendix D show both, but the main text seems ambiguous).
2. Proposition 5.1 relies on the assumption in the proof that $\epsilon < 1/2$ (e.g., for DPO hard labels $\epsilon=0$). Given that CDPO and IPO use soft labels where $\epsilon$ is non-zero, this should be explicitly stated as a condition for the proposition's claim ($\pi_{\theta^*}(y_w|x) > \pi_{ref}(y_w|x)$) to hold, or confirm if the result holds for all $\epsilon$.

---

> ### Author Response · Authors · 2025-11-28
> **Response to reviewer vUaj**
>
> We sincerely thank the reviewer for their positive evaluation and for recognizing the value of our theoretical contributions and practical improvements. We appreciate the careful reading and the specific technical feedback.
>
> ***The paper introduces $l_1$ and $l_2$ penalties for the regularization term in Section 5.1, but the loss equations (12 and 15) only show the squared penalty.***
>
> The reviewer correctly notes that "the loss equations only show the squared penalty" while we evaluate both $l_1$ and $l_2$ variants.
> We have clarified this in section 5 in the revision.
>
> ***Proposition 5.1 relies on the assumption in the proof that $\epsilon<1/2$.***
>
> We agree and have revised Proposition 5.1 to explicitly state the condition.
>
> We mark the associated changes in the revised paper in red color. We may not have highlighted other minor adjustments that helped with the natural incorporation of the requested changes.

---

### Review · Reviewer_Yjk2 · 2025-11-22

**Summary Of Contributions:**

The paper shows that the standard DPO loss can also be derived from an alternative RL-style objective with a KL penalty applied only to a subset of responses, and uses this formulation to prove conditions under which all labeled responses probabilities are systematically shrunk ("likelihood displacement").
It then presents a unified classification view of several DPO-style methods, arguing these pairwise objectives are under-specified because they only constrain probability ratios within each pair.
To mitigate this, the authors introduce C2-DPO, a family of DPO variants with simple pairwise constraints (e.g., preserving the sum of log-probs or probabilities of winner+loser) that provably enforce "winner higher, loser lower" relative to the reference and yield modest empirical improvements over vanilla DPO and a few recent baselines.

**Audience:**

Yes

**Audience Explanation:**

The derivation of the DPO loss from an alternative RL objective where the KL penalty is only on in-sample responses and Section 4 "A Classification View of DPO-style Algorithms" would be interesting and useful for some individuals in TMLR's audience.

**Claims And Evidence:**

Yes

**Claims Explanation:**

Overall, most of the central technical claims are supported by correct derivations and reasonably chosen experiments, but a few parts of the narrative feel stronger than what the evidence strictly justifies.

For example, "both the preferred and rejected responses tend to decrease in probability under its optimal policy" is the main motivation to add the regularization/constraints.
However, the formal result (Lemma 3.2) shows that if a labeled response has lower reward than some unlabeled response, then at the optimum of the in-subset KL problem its probability shrinks by at least a factor $e^{-1}$.
This is a conditional, per-pair statement; the upgrade to global language like "tend to decrease" for both responses relies on informal plausibility arguments and a few illustrative plots, rather than a quantitative probabilistic analysis of how often the lemma's condition actually holds.

Empirical claims about "higher-quality final models" are not sufficiently supported. The experiments do show modest, consistent gains over vanilla DPO and a few recent DPO-style baselines on UltraFeedback, TL;DR, and MT-Bench for a small set of models.
However, these are relatively narrow setups, and there is no comparison to very simple practitioner baselines (e.g., DPO + explicit KL regularization), nor robustness checks across seeds or evaluators.
So the broad wording about "delivering higher-quality final models" overstates what the current empirical evidence can support.

**Requested Changes:**

## Clarify and abstract the mathematical core

I strongly recommend restructuring Sections 3 to first present the underlying optimization problems and constraints in a fully abstract setting without heavy RLHF/DPO jargon and only then reintroduce the RLHF/DPO interpretation.
This would make the main contributions (closed-form solutions, under-specification, constraints) much clearer.

The current exposition often mixes informal phrases ("following the recipe…", "in-sample prompt–response pair") with mathematically non-essential application details, which can obscure the structure of the arguments.
Separating definitions from narrative and clearly distinguishing the abstract math from the RLHF interpretation would make the paper easier to read and the contributions easier to evaluate.

## Make Lemma 3.1 and related derivations more precise

Lemma 3.1 currently states the existence of a "closed-form solution" but only uses a difference identity and a reference to the original DPO derivation.
I would like to see the solution of Eq. (5) stated explicitly (with the relevant case distinction from Lemma A.1) and the steps from that solution to the DPO loss written out clearly. This would also help make transparent which properties of the underlying RL objective are preserved (or discarded) when passing to the DPO surrogate.

## Connect more explicitly to learning-to-rank and preference-learning literature (non-critical)

Much of the conceptual content (e.g., pairwise Bradley–Terry modeling, non-identifiability of scores under ranking-equivalent transformations, and the need for additional regularization/constraints), is well known in the learning-to-rank and preference-learning communities.
A more explicit discussion of how DPO-style methods fit into that landscape, and what is genuinely new in the RLHF-specific setting would significantly strengthen the positioning.

---

> ### Author Response · Authors · 2025-11-28
> **Response to reviewer Yjk2**
>
> We thank the reviewer for their thorough evaluation and highly constructive feedback. Below we address the main concerns.
>
> ***The claim about probability reduction "relies on informal plausibility arguments and a few illustrative plots, rather than a quantitative probabilistic analysis."***
>
> We appreciate this observation. We note that the condition in Lemma 3.2 requires in-sample rewards to be dominated by out-of-sample rewards. Since the in-sample set typically contains few responses while the out-of-sample space is effectively infinite for language models, this condition is plausible. Figure 2 (Right) provides empirical support:\, where the sum of log-ratios decrease substantially during training across all checkpoints. Additionally, likelihood displacement has been independently observed in numerous recent studies (see section 3 for a detailed list). Our contribution is providing a theoretical explanation for this widely-observed behavior via the subset-constrained KL formulation. We have revised the language to be more careful about the scope of our claims.
>
> ***There is no comparison to very simple practitioner baselines (e.g., DPO + explicit KL regularization), nor robustness checks across seeds or evaluators.***
>
> Regarding DPO with explicit KL: Figure 2 (Left) demonstrates that increasing $\beta$ in DPO controls the KL divergence between the learned and reference policies. This shows the implicit regularization is effective. The baseline of DPO with explicit KL would require sampling from the policy during training to estimate the KL divergence, adding significant computational cost. In contrast, C2-DPO operates on in-sample responses only, requiring no sampling.
>
> Regarding robustness checks: Our experiments include multiple forms of validation. We report win rates with standard errors over $10$ inference runs to demonstrate stability. For evaluation, we use both Claude 3.5 Sonnet for pairwise comparisons and MT-Bench which employs GPT-4 as judge across $8$ different task categories, providing multi-evaluator validation.
>
> ***Addressing requested changes***
>
> We implemented all three requested changes in the revision. Section 3 now begins with the optimization framework before introducing the RLHF interpretation. We made Lemma 3.1 more precise by adding the detailed steps from the closed-form solution to the DPO loss in the proof, and by adding Remark 3.1 which explicitly states the solution with case distinction. We also added discussion at the end of section 7 on connecting to the learning-to-rank literature while clarifying what is genuinely new in the RLHF setting.
>
> We mark the associated changes in the revised paper in red color. We may not have highlighted other minor adjustments that helped with the natural incorporation of the requested changes.

---

### Review · Reviewer_DN7o · 2025-11-23

**Summary Of Contributions:**

This work studies the alignment problem and presents several theoretical results. It begins with reviewing direct preference optimization (DPO), a mainstream algorithm for solving the alignment problem in reinforcement learning from human feedback (RLHF). The paper presents an alternative form of DPO derived by limiting the KL-divergence term to in-sample responses rather than to the full output distribution. The proposal of this alternative form leads to some counter-intuitive implications, including a decrease in the likelihood of both preferred and rejected responses under the optimal policy.. Next, the paper introduces Constrained Controlled DPO (C2-DPO) based on these theoretical analyses and provides experimental results that demonstrate its efficacy on several datasets.

**Audience:**

Yes

**Audience Explanation:**

Please see my answer above.

**Broader Impact Concerns:**

None.

**Claims And Evidence:**

Yes

**Claims Explanation:**

This paper is not in my research field, and I have no prior experience with RLHF or any related topics. Therefore, I will focus on my high-level understanding of this work and leave its technical rigor and merits to more experienced reviewers.

The paper is well written. I like the way the authors organized the technical sections. Even as a novice in this area, I can follow the high-level story thanks to the paper's writing.

The paper builds a theoretical framework for analyzing DPO and reveals some counter-intuitive implications. The theoretical analysis developed in this work explains existing DPO algorithms from an interesting and novel (to my knowledge) perspective. It opens new opportunities and may inspire follow-up work to conduct similar theoretical analyses in this direction.

For example, the idea of unifying several existing DPO-style algorithms under the same classification framework (Sec. 4), loaded with different objectives and data labels, seems to be a useful interpretation of these existing algorithms. I appreciate this effort.

Overall, I am inclined to answer Yes to this question, based on what I have read from the paper so far. However, I have to stress that I am not a researcher in this field, and I request that the editor and other reviewers take this factor into account.

**Requested Changes:**

I have a small request for clarifying the technical exposition in the experimental results: I can read from the tables and figures that the proposed C2-DPO leads to improvements over prior works, but I do not have sufficient background information to judge whether such improvements are marginal or significant, particularly for those blue-red (DPO vs. C2-DPO) histograms in Figs. 4, 6, and 8. Again, this is primarily because this paper is not in my research field, and I very much welcome the authors' providing more context behind these numbers.

---

> ### Author Response · Authors · 2025-11-28
> **Response to reviewer DN7o**
>
> We sincerely thank the reviewer for their thoughtful evaluation and positive assessment of our theoretical contributions and exposition. We appreciate the recognition on the alternative DPO derivation.
>
> ***Contexts for the empirical improvements***
>
> The reviewer requests more context behind these numbers to better judge the improvements, particularly in Figures 4, 6, and 8.
> We appreciate this question and will add context in the revision. The improvements shown are modest but meaningful in the alignment literature. Specifically:
> Win rate improvements of $2-5\%$ (as shown in the histograms) are in a similar range as many recent DPO variants. The alignment community often prioritizes methods that provide consistent gains across diverse evaluation settings over those showing large but unstable improvements.
> MT-Bench scores (radar plots in Figures 5 and 7) show C2-DPO outperforms baselines across multiple task categories. MT-Bench uses GPT-4 to evaluate model responses across 8 domains. Consistent improvements in all categories, as we observe, represent meaningful quality gains.
> Head-to-head comparisons (Figure 4) show C2-DPO wins against vanilla DPO in $55-60\%$ of cases. In preference learning, where even state-of-the-art methods often perform comparably, consistent advantages above $50\%$ baseline indicate real improvements.
>
> We revised Section 6 to provide more context to help readers interpret the results. We acknowledge these are not dramatic improvements, but they are achieved with minimal computational overhead and theoretical guarantees about controlling likelihood displacement.
>
> We mark the associated changes in the revised paper in red color. We may not have highlighted other minor adjustments that helped with the natural incorporation of the required changes.

---

### Decision · Action_Editor_Amyb · 2026-01-17

**Recommendation:** Accept as is

**Audience:**

Yes

**Audience Explanation:**

Researchers/practitioners using DPO/RLHF caring about a plausible explanation for why DPO can shift likelihood mass in unintuitive ways. The proposed fix is simple to implement (no extra sampling) and appears to improve outcomes across common benchmarks.

**Claims And Evidence:**

Yes

**Claims Explanation:**

The paper shows DPO can be seen as coming from an “in-sample only” KL-regularized objective (rather than KL over the whole output distribution). This helps theoretically justify the empirical effect where both the preferred and rejected responses can lose probability (referred to as “likelihood displacement” in previous literature). Given that KL is computed on the subset of in-sample responses, it can become negative if \pi_theta(y) < \pi_ref(y) thus maximizing the objective (5) when \pi_theta(y) decreases for both preferred and non-preferred responses; Lemma 3.2 shows makes this formal by showing that in-sample responses (preferred or dispreferred) "experience probability reduction whenever their rewards are dominated by out-of-sample rewards". The authors propose a fix C2-DPO which constrain the prob of preferred and dispreferred to move in opposite directions. Experiments generally support “C2-DPO improves over DPO” across the reported settings, although I wouldn't say it's the most important point of the paper. Formal claims are accurate. Papers give a formal explanation to an empirical observed behavior.